# Shrinking the footprint of the criminal legal system through policies informed by psychology and neuroscience

Arielle Baskin-Sommers [1,2] ✉, Alex Williams[1], Callie Benson-Williams[1], Sonia Ruiz[1], Jordyn R. Ricard [1] & Jorge Camacho[2]

The footprint of the legal system in the United States is expansive. Applying psychological and neuroscience research to understand or predict individual criminal behavior is problematic. Nonetheless, psychology and neuroscience can contribute substantially to the betterment of the criminal legal system and the outcomes it produces. We argue that scientific findings should be applied to the legal system through systemwide policy changes. Specifically, we discuss how science can shape policies around pollution in prisons, the use of solitary confinement, and the law's conceptualization of insanity. Policies informed by psychology and neuroscience have the potential to affect meaningful—and much-needed—legal change.

On any given day there are more than 1.9 million people behind bars in jails or prisons in the United States[1]. Nearly half of all adults living in the United States experience incarceration in their family[2]. Most who encounter the criminal legal system are dealing with problems related to poverty and mental illness, which worsen with arrest and incarceration[2–5].

With the hope of trying to shrink the footprint of the criminal legal system on American families, over the past two decades, much discussion has focused on the applicability of psychology and neuroscience to the legal system. These discussions are rife with conjecture around the notion that psychology and neuroscience can detect liars, objectively determine criminal responsibility, and predict who will engage in violent behavior. Unfortunately, the framing of psychology and neuroscience as being able to transform the law by focusing on the individual reflects a misrepresentation of the science and the standards of law.

Psychology and neuroscience provide probabilistic, not deterministic, estimates of phenomena in the aggregate. While psychological and neuroscientific findings may be valid for a given group in general, they may not apply to a particular individual within that group (often referred to as the group-to-individual problem). Thus, psychological and neuroscientific techniques cannot show beyond a reasonable doubt that distinct brain structures or abnormalities affect the mental state of a particular individual at the time of the crime, that they will undoubtedly engage in criminal conduct in the future, or evidence of mitigation at the sentencing phase above and beyond other less expensive and more reliable tools (e.g., assessing family history or exposure to violence).

While there is much skepticism about the use of psychology and neuroscience in the legal system, these disciplines do have the potential to affect meaningful change in how the legal system operates and in the outcomes it produces. In this perspective piece, we will argue that psychological and neuroscientific findings can be applied to and improve aspects of the legal system through policy changes. We will focus on how science can shape policies that affect those who are incarcerated in jails and prisons, and by extension society at large. There is a substantial body of research delineating the negative impact of incarceration on individuals (e.g., negative effects on health, mental health, job prospects, educational attainment, etc.)[6–9] and their families[2,10]. Here, we select three aspects of where and who is incarcerated and detail how policies surrounding these aspects can or should be influenced by emerging findings in psychology and neuroscience. Specifically, we highlight how the issues of pollution in prisons, the use of solitary confinement, and the restrictions of the legal concept of insanity could be reshaped by integrating scientific findings.

## Criminal legal system aspects of interest
### Pollution: toxins and noise

The United States continues to incarcerate more people than any other country. Over 6000 facilities hold almost 2 million people. The long reach of incarceration substantially reduces the chances of a formerly incarcerated person obtaining an education, stable employment, owning a home, or living above the poverty line[5]. Further, exposure to toxins and noise pollution within jails and prisons in the United States will likely have substantial negative effects on the individual's psychological and brain health.

[1]Department of Psychology, Yale University, 100 College St, New Haven, CT 06510, USA. [2]Yale Law School, 127 Wall St, New Haven, CT 06511, USA.
✉e-mail: arielle.baskin-sommers@yale.edu

There are documented violations, ranging from inadequate sewage and waste disposal to poor water quality and the presence of toxins such as asbestos, manganese, and lead, in jails and prisons throughout the United States[11–16]. For example, since 2000, over a quarter of California's state prisons have been cited for major water pollution problems[13]. Rikers Island, a jail in New York City, was built atop a toxic landfill in 1932[17,18] that in 2011 the New York City Department of Correction reported was still emitting poisonous gases[19]. Since 2020, at least 23 jails have been either proposed or constructed on toxic and contaminated lands[16]. Further, regulations that would protect the general population against toxin exposure often are not in place for jails and prisons (e.g., the Environmental Protection Agency designated that most parts of prisons and juvenile detention centers are zero-bedroom dwellings [i.e., residential dwellings where living areas are combined with sleeping areas] and therefore are not subject to the Lead Renovation, Repair and Painting Rule)[20]. Exposure to such toxins causes health problems, including cancer, hypertension, and neurodegeneration, as well as mental health problems, including impulsivity and aggression[21,22].

Similarly, noise pollution is an issue in jails and prisons[23]. Sources of noise in prisons are unpredictable and come from multiple streams. These facilities often are built using hard, reflective materials that heighten noise pollution. The U.S. Environmental Protection Agency defines acceptable levels of noise in residential areas, hospitals, and schools as 45 dB(A)[24]. However, the American Correctional Association set noise standards for correctional housing to not exceed 70 dB(A)[25]. Long-term exposure to sound above 50 dB(A) has been shown to cause serious health issues, such as increases in stress hormones, cardiac problems, and hypertension[26,27].

Research in psychology and neuroscience provides key findings that support the claim that exposure to toxins and noise in prisons can negatively impact physical and mental health. With regard to toxins, research in non-human animals and humans shows that exposure to chemicals such as lead, arsenic, and manganese cause serious harm. Specifically, documented harms include damage to dopaminergic neurons (which regulate motivation, reward, and habit learning[28]) and increase beta-amyloid protein plaques and intracellular neurofibrillary tangles (which characterize Alzheimer's Disease)[29]. Additionally, exposure to such toxins result in deficits in the structure and function of the hippocampus (a region of the brain important for memory and learning[30]), increase neuroinflammation, and produce general poorer brain health[31–35]. Furthermore, high concentrations of neurotoxic chemicals and persistent pollutants have an undisputed impact on cognition and are associated with deficits in general cognitive functioning, IQ, executive functioning, language, and memory[21,22,36,37]. Of utmost relevance for the legal system, toxin exposure in the short-to-mid-term is linked to heightened levels of impulsivity, hyperactivity, and aggressive behaviors[11,38–41].

Noise pollution and chronic noise exposure also have long been considered an ecological stressors that impact psychological and neural functioning. Prolonged noise exposure causes clinically impairing distress and stress hormone dysregulation[27]. Studies with non-human animals and humans link chronic noise exposure, particularly unpredictable noise, to damage to the central nervous system, the generation of pathological neurofibrillary tangles (which is related to Alzheimer's disease), and poorer tissue health in the hippocampus, prefrontal cortex (a region related to self-control), and amygdala (a region important for emotion processing and regulation)[42–46]. These neural alterations appear to persist even after noise exposure stops, suggesting both short- and long-term neurological impacts due to chronic noise exposure.

There are clear connections between pollution, toxin and noise, and physical and mental health problems. These pollutants have the potential to negatively impact neural regions responsible for basic emotion, cognition, and behavioral control. Using findings from psychology and neuroscience to understand the effects of toxin and noise pollution across species necessitates improvements in the ecology of jails and prisons.

A significant problem with current jail/prison environmental policies lies in the oversight of facilities and the enforcement, or lack thereof, of policies intended to ensure environmental safety. Frequently, jail and prison facilities are constructed in areas where significant ecological risk factors exist and require substantial remediation efforts to ensure safe occupancy, but these efforts either fail to materialize or are abandoned before completion[15,47]. The result has been exposure and vulnerability to serious health and safety risk factors like toxins or ecological disaster[48]. The failure to complete mandated remediation can be compounded by reduced access to legal remedies by incarcerated populations[49].

To shrink the footprint of this aspect of incarceration, policymakers should prioritize two strategies. First, they should redouble their efforts to enforce existing laws and regulations that govern applicable environmental standards and ensure that remediation efforts are completed. Second, they should adopt a principle that no policy that limits movement, fraternization, occupational activities or contact with outside environments/persons should be issued without an evidence-based accounting of the harms associated with that policy, including strategies for addressing those harms[50]. With sufficient will and attention to these problems, there is reason to believe that conditions and outcomes within jails and prisons can be substantially improved.

## Solitary confinement

Solitary confinement refers to the physical and social isolation of an individual in a cell for twenty-two to twenty-four hours a day. The cells typically are sparse, consisting of a steel door, a bed, a toilet, and a sink. Loud, unpredictable noise permeates the space that is no bigger than 6 feet x 9 feet[51], and many cells lack natural light. People are in solitary confinement for periods that range from days to weeks, months, years, or even decades[51]. In 2021, approximately 48,000 individuals were held in solitary confinement[51]. Ten percent of people in solitary had been held for three years[51]. One may reasonably presume that the severity of solitary confinement would tend toward its sparing use, reserved only for the most egregious and dangerous offenders. However, the reality is that people can be placed in solitary confinement for various reasons, including for minor disciplinary infractions or for their safety[52]. The latter holds true for those deemed to be particularly vulnerable to victimization within incarcerated populations, including LGBTQIA persons, pregnant persons, and those with mental illness[51]. Although isolation for one's protection can be voluntarily requested by an incarcerated person, jails and prisons can exercise their discretion to involuntarily isolate someone when officials determine that they cannot otherwise ensure that person's safety, resulting in involuntary confinement that is largely indistinguishable from more punitively-motivated solitary confinement.

Research on solitary confinement includes qualitative accounts of incarcerated persons' experiences and empirical studies examining the relationship between this aspect of incarceration and safety, mental health, and criminogenic risk. While the qualitative accounts, as well as popular media sources and theory-based writings from scholars, document the harrowing effects of solitary on individuals[53,54], the empirical evidence supporting the negative effects of solitary on safety, mental health and criminogenic risk is more mixed. Some studies fail to detect effects of solitary confinement on individual behavior and mental health[55–58]. Other studies document significant negative effects of solitary on incarcerated people's physical and mental health[59–68], particularly in terms of anxiety, psychotic symptoms, sensory arousal, and behaviors that effect mortality by any or unnatural causes (e.g., suicide)[57,64]. Additionally, there is evidence that being housed in solitary confinement, even for a week, can change alpha frequencies measured by EEG[57,69]. The U.S. Department of Justice acknowledges that solitary confinement can worsen existing mental illnesses and trigger new ones[70].

The study of solitary confinement is understandably very difficult. Some studies cited above lack appropriate methodological controls (e.g., randomization, comparison groups), were conducted in small samples, and/or were the result of litigation possibly introducing bias into the method[71,72]. Unequivocal empirical evidence for concluding that the practice of solitary confinement in jails and prisons is uniformly negative is lacking, leading some scholars[55,57] to suggest caution in developing policy based on an

incomplete science. However, there is a more substantial evidence base on the negative effects of solitary conditions in research with non-human animals and humans outside of the jail/prison context. While research in laboratories or in other institutional setting is not identical to incarceration-based solitary, there is a strong basis for comparing the effects of physical isolation and the deprivation of basic experiences.

Numerous studies with non-human animals explore what happens to the brain and behavior when subjects are physically isolated, deprived of resources, and are deprived of sensory information. These studies document trends including the expression of hyperactivity, altered responses to stressors, cognitive impairments, increased aggression, and alterations in mesolimbic dopamine functioning (which is important for learning and goal-directed behaviors)[73-75]. Rats in isolation also experienced lasting changes in psychological (e.g., aggression or fear of new situations), cognitive (e.g., declines in mental flexibility), and neural (e.g., reduced prefrontal cortex volume, decreased cortical and hippocampal synaptic plasticity, or alteration in the mesolimbic dopaminergic system) functioning as compared to rats in stimulating or complex environments[76-81].

Similar patterns are found in some human studies, particularly those involving youth exposed to institutional settings characterized by deprivation of interpersonal contact. In one longitudinal and randomized study of children monitored through the Bucharest Early Intervention Project (https://www.bucharestearlyinterventionproject.org/about-beip), youth with histories of institutional residence had indicators of significantly worse brain health and atypicalities in neural structure, function, and communication compared to non-institutionalized youth[82-86]. Further, youth experiencing psychosocial deprivation display deficits in memory and executive functions compared to non-institutionalized youth[87,88]. The randomized design of the Bucharest Early Intervention Project provides some of the strongest causal evidence of the impact of isolation on development, with lasting effects.

Together, extant non-human and human research serve as evidence that psychological and neural differences are either generated or exacerbated by conditions of isolation. Solitary is not only painful in itself but also "undermines people's sense of belonging, control, self-esteem, and meaningfulness … reduces pro-social behavior, and impairs self-regulation"[89]. Research across disciplines, then, provides a clear foundation that, on average, solitary confinement or similar conditions is physically and psychologically harmful.

In 2016, President Obama adopted a recommendation to end solitary confinement for juveniles in federal prisons. However, in 2023, 11 states still have no limits on the use of solitary confinement for juveniles, and just under half the states have passed laws that narrow the use of solitary confinement in juvenile facilities[90]. In 2023, the U.S. House of Representatives introduced a bill to ban solitary confinement in federal prisons[91]. To date, however, similar bills have not passed.

The footprint of solitary confinement, including deleterious psychological and neural effects (above and beyond just incarceration), has been argued in the courts to represent an Eighth Amendment violation that constitutes cruel and unusual punishment (see arguments from *Ashker v. Brown*)[92,93]. Solitary confinement should be used only for brief periods and as a very last resort. The United Nations Standard Minimum Rules for the Treatment of Prisoners[94]— known as the Mandela Rules—condemn the use of solitary for people with mental and physical disabilities; such rules should be mandated in the United States across federal and state levels. They would serve to protect not only the incarcerated individual, but also the facility staff and society at large.

### Redefining the legal concept of insanity

The U.S. legal system is continuously confronted with the need to adjudicate, assess, and treat people with mental illness[95-97]. How the law defines mental illness can have a substantial impact on how individuals who enter the system are judged and handled. For instance, in the United States, prevailing legal doctrines, including under the Model Penal Code, which has been adopted by 20 states, dictate that individuals may be considered less

responsible if they can show that "at the time [their criminal conduct was] a result of mental disease or defect" indicating that the person "lacks substantial capacity either to appreciate the criminality [wrongfulness] of [their] conduct or to conform [their] conduct to the requirements of law"[98-100], and therefore they can be found not guilty by reason of insanity. A successful determination of not guilty by reason of insanity can then trigger a therapeutic intervention via placement in a forensic mental health center (i.e., justice-involved treatment setting) over a punitive intervention via incarceration in a traditional prison.

However, the insanity defense is rarely used in practice because it is very difficult to demonstrate legal insanity[101]. Additionally, some legal policies greatly limit who even qualifies to present this defense. For example, the Model Penal Code's insanity defense excludes disorders characterized by repeated criminal or antisocial conduct. Here, we argue that the disconnect between legal conceptualizations of insanity on one end and psychological and neuroscientific understandings on the other can lead to the inadequate acknowledgment of many mental health problems in the criminal process.

One of the difficulties in referring to insanity in legal proceedings is the disconnect between terms used in the law and how they would be considered in psychology/neuroscience. For example, legal policies related to insanity refer to "mental defect" or "defect of reason" as a premise for questioning criminal responsibility[98]. In the law, there is no clear definition of what is meant by these specific phrases. In psychology and neuroscience, we might operationalize these phrases as an aberration in cognition and emotion that undermine accurate perception, interpretation, and/or reaction to information. This operationalization provides a biopsychology basis for understanding an individual's conduct[102]. As another example, "disease of the mind" is noted in some insanity doctrines[98], again without a clear definition. In psychology and neuroscience, we might operationalize this phrase as brain-based pathology resulting from various causes (e.g., injury, genetics, environmental stress) and that is characterized by identifiable signs or symptoms. In this case, a biopsychological definition would specify the type of evidence needed to initiate a defense based on insanity. As a result of bridging the gap between the language of the law and science, individuals with disorders where psychological and neuroscientific evidence provides a clear basis for disruptions that undermine cognition, affect, and behavior should[103,104], without question, be eligible to put an insanity defense. However, the lack of a clear, objective, evidence-informed legal standard for identifying insanity precludes this outcome.

A shift in the legal policy around insanity would provide a scientific-based basis for determining the groups of people who are eligible for such a defense. It is then up to courts to determine if there is clear evidence that the specific factors played a role in an individual's behavior. At this time, though, the courts cannot properly make these determinations without the ability to conduct a frank assessment of any intersectionality between mental illness and criminality. Unfortunately, the prevailing legal standards around insanity preclude these very assessments based on ill-defined terminology and exclusion of certain disorders. By widening the potential eligibility for an insanity defense based on scientific evidence, many people currently ensnared in the legal system may qualify for special protections under the law and might need to be mandated to treatment. Further, psychological treatments that specifically target the neural basis of these cognitive and affective psychological differences already exist, such as cognitive training programs that target attentional/other cognitive biases, emotion regulation strategies, or behavioral treatments that target reward hypersensitivity[103,105-108], providing an opportunity for rehabilitation. Broadening the scope of individuals who may be eligible for consideration under insanity doctrines could drastically reshape how mental illnesses are handled in the legal system, perhaps reducing the current footprint of a punitive system and shifting the focus to a system that more properly considers the role of mental health problems in some people's behavior. If done correctly, this shift should feasibly improve safety outcomes, both individually and systemically, through deliberate intervention against underlying psychological motivators of behavior.

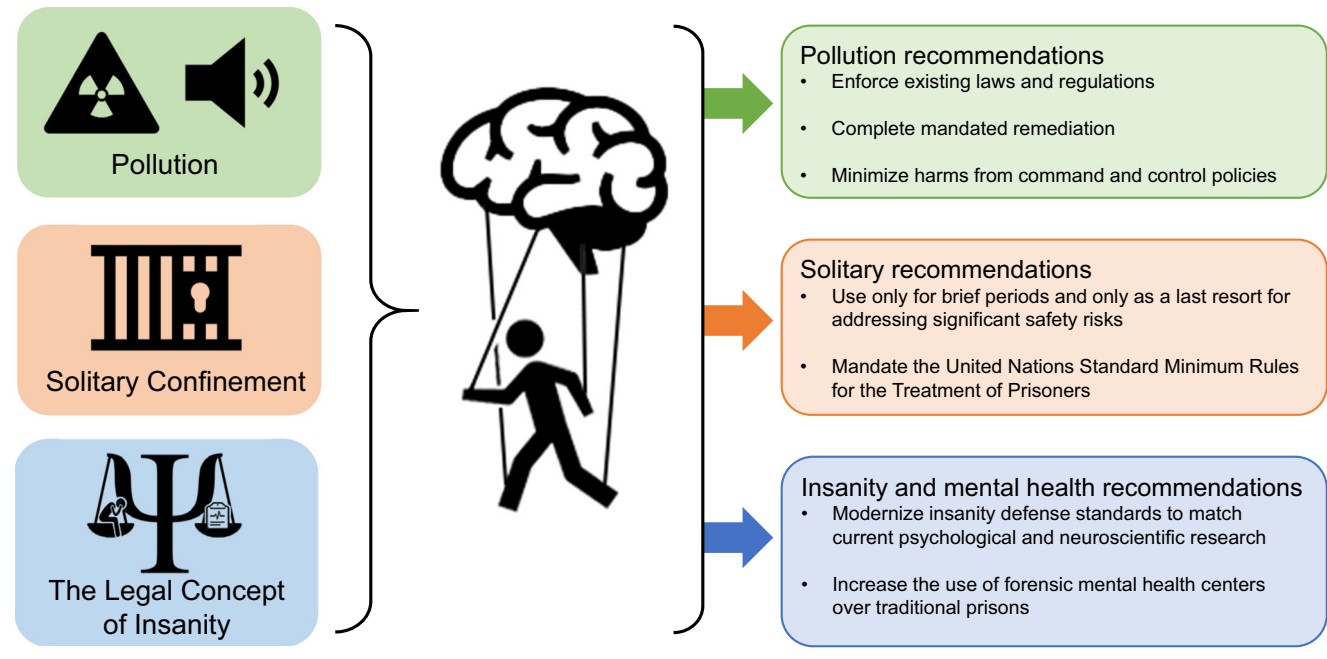

**Fig. 1 | Using psychology and neuroscience research to inform criminal legal policy.** Summary of criminal legal system aspects of interest and policy recommendations based on psychology and neuroscience research.

## Outlook

Psychological and neuroscientific findings are compelling as they apply to the impact of pollution and solitary confinement on behavior and the brain. Psychological and neuroscientific findings that challenge our understanding of 'insanity' raise questions about the handling of mental health problems in the current legal structures. Using research grounded in psychology and neuroscience in each of these aspects of the legal system overcomes some of the limitations outlined above with regard to the ecological fallacies and deterministic assumptions often made when applying evidence to the criminal legal system--instead of focusing on the individual, we can apply science to inform policy changes that affect groups of individuals (see Fig. 1 for summary).

In a landscape that often looks plagued by injustice, lacks an empirical evidence base, and imposes a tremendous cost on individuals and society both in terms of crime and punishment, it is imperative to look for alternative ways of integrating psychology and neuroscience findings and improving policies. If implemented appropriately, these robust psychological and neuroscientific findings have the tremendous potential to affect meaningful—and much-needed—legal change in the United States today.

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

## Acknowledgements

We thank the Justice Collaboratory at Yale Law School for providing an interdisciplinary scholarly environment where these ideas can grow.

## Author contributions

A.B.S. and J.C. conceptualized the manuscript. A.B.S., A.W., C.B.W. and J.C. conducted literature searches. A.B.S. and J.C. wrote the initial draft. A.B.S., A.W., C.B.W., S.R. and J.R.R., J.C. edited and approved the final text.

## Competing interests

The authors declare no competing interests.
