## [Peer Review File · Communications Psychology]

11th Oct 23

Dear Arielle,

Thank you for your patience during the peer-review process. Your manuscript titled "Shrinking the Footprint of the Criminal Legal System through Policies Informed by Psychology and Neuroscience" has now been seen by 2 reviewers, and I include their comments at the end of this message.

The reviewers are in principle enthusiastic about your work. However, they also mention a number of concerns. We are interested in the possibility of publishing your manuscript in *Communications Psychology*, but would like to consider your response to these concerns in the form of a significantly revised manuscript before we make a decision on publication.

A Perspective has two key elements: a fair literature review and a - potentially opinionated - proposal that arises from the authors' reading of this literature. Although the general character of the piece is therefore more opinion-focused than that of a Review article, the literature review must be balanced, fair, and comprehensive.

Both Reviewers, and especially Reviewer #1 highlight instances in which the literature review lacks persuasiveness because it appears biased or incomplete. The key to a successful revision of your manuscript will be to improve (extend and balance) the literature review, and add to that a compelling proposal for the change in the criminal justice system you consider warranted. You may also wish to extend your outlook to accommodate Reviewer #2's suggestion about how the criminal justice system might be improved to serve people with mental illness.

In sum, we invite you to revise your manuscript taking into account all reviewer and editor comments.

EDITORIAL POLICIES AND FORMATTING

You will find a complete list of formatting requirements following this link:
<https://www.nature.com/documents/commsj-style-formatting-checklist-review-perspective.pdf>
Please use the checklist to prepare your manuscript for resubmission.

* **TRANSPARENT PEER REVIEW:** *Communications Psychology* uses a transparent peer review system. This means that we publish the editorial decision letters including Reviewers' comments to the authors and the author rebuttal letters online as a supplementary peer review file. We publish these records for all accepted manuscripts. However, on author request, confidential information and data can be removed from the published reviewer reports and rebuttal letters prior to publication. If your manuscript has been previously reviewed at another journal, those Reviewers' comments would not form part of the published peer review file.

If you have any questions about any of our policies or formatting, please don't hesitate to contact me.

Please use the following link to submit your revised manuscript and a point-by-point response to the referees' comments (which should be in a separate document to any cover letter):
[Link redacted]

** This url links to your confidential home page and associated information about manuscripts you may have submitted or be reviewing for us. If you wish to forward this email to co-authors, please

delete the link to your homepage first **

We hope to receive your revised paper within 12 weeks; please let us know if you aren't able to submit it within this time so that we can discuss how best to proceed. If we don't hear from you, and the revision process takes significantly longer, we may close your file.

We understand that due to the current global situation, the time required for revision may be longer than usual. We would appreciate it if you could keep us informed about an estimated timescale for resubmission, to facilitate our planning. Of course, if you are unable to estimate, we are happy to accommodate necessary extensions nevertheless.

Please do not hesitate to contact me if you have any questions or would like to discuss these revisions further. We look forward to seeing the revised manuscript and thank you for the opportunity to review your work.

Best,
Marika

Marika Schiffer, PhD
Chief Editor
Communications Psychology

REVIEWERS' EXPERTISE:

Reviewer #1: criminal justice system, psychiatry/psychology
Reviewer #2 criminal justice system, psychiatry/psychology

REVIEWERS' COMMENTS:

Reviewer #1 (Remarks to the Author):

The purpose of this commentary is to discuss deleterious impact of the criminal legal system and how psychology and neuroscience can reduce the footprint and consequences of the expansive criminal legal system in the United States. This is an interesting paper highlighting an important issue that is understudied and under-discussed (I've worked in corrections and been a correctional researcher for nearly 25 years and I cannot recall a paper I've read on the criminal legal footprint). The use of psychology and neuroscience certainly offers novel opportunities for combatting the impact of this footprint.

The strengths of this manuscript lie in the novelty of the topic and discussion of science, including neuroscience, to combat the negative impact of the criminal legal footprint. Limitations of the manuscript include instances of overstating results to make a point, and a lack of objectivity throughout the paper. I do see potential in this manuscript, but for this commentary to contribute to the scientific literature, a more objective and neutral position is warranted. Further, the direct recommendations and implications of how psychological science and neuroscience can positively alter current directions needs further development. As presented, the recommendations are overly vague and non-descript; thus, limiting the impact of the conclusions and recommendations. Specific suggestions for improving the manuscript are provided below:

1. On page 1, the authors write, "In most jails and prisons, there are well-documented violations..." I

appreciate the citations, but does the evidence support this conclusion for "most jails and prisons?" Certainly, problems exist, but I'm not aware that the prevalence has been established and that it extends to most of the thousands of jails and prisons across the United States. In fact, the paper goes on to counter this conclusion by citing that ¼ of California prisons have water contamination problems. This is disconcerting for sure, but does not represent "most" of the prisons in California, in fact, less than one-half. A more neutral position throughout the paper would increase the appearance of objectivity, and, thus, confidence in the conclusions.

2. The review of impacts of incarceration, and all its variables, is superficial (p. 2-3). There is a body of work examining the impacts of incarceration on persons with mental illness that would allow for a more thorough discussion of the impact of jail/prison on individuals mental health functioning (including policy, culture, and environmental impacts).

3. The discussion of solitary confinement is also superficial and does not take into account studies that demonstrate negligible effects. The literature reviewed is not inaccurate, but limiting the review to one perspective while ignoring a counter-perspective is incomplete at best, and biased at worst. The relevant points can be made with a more neutral review of the scientific literature.

4. The use of psychopathy as the primary example of treating and managing persons with mental illness is problematic. Psychopathy is a personality disorder, and is clinically different from persons with severe mental illness (which represents, on average, 10-15% of the prison population, where as the prevalence rates for personality disorders are substantially higher). A better example would be a person with schizophrenia or bipolar disorder in jails and prisons (not insanity acquittals as they are patients in the mental health systems, not criminal legal systems). It would also be relevant to note, that the over-representation of people with mental illness is not merely a result of the deinstitutionalization movement as a relatively robust body of research demonstrates that justice-involved persons with mental illness present with criminal risk comparable to their non-mentally ill peers in the criminal legal system (see works by Drs. Jim Bonta, Jennifer Skeem, Jeffrey Drain, Robert Morgan, et al.).

5. The opinion that persons with psychopathy qualify for the insanity defense is not a policy or practice issue, it is a legal determination defined in federal and state statutes. This opinion is also an outlier opinion not well supported in the paper and is unlikely to be supported by the majority of mental health, legal, correctional or policy professionals that work in the criminal legal system. Again, an important concept not adequately covered in this opinion regarding psychopathy specifically and mental illness broadly is the prevalence of criminal risk.

Reviewer #2 (Remarks to the Author):

This is an interesting piece that turns the common narrative about psychology in the law (e.g., that psychologists can detect liars or predict crimes) into ways that psychological findings can be used to decrease harms caused by incarceration. The authors' recommendations are (or should be) common sense -- improve the quality of the jail/prison environment; reduce use of solitary confinement; revisit the definition of "insanity" to encompass a broader range of psychopathology. However, the piece does an impressive job of bringing together a concise summary of current and strong evidence from the literature to support each recommendation. This is more compelling than the argument that it's "just the right thing to do."

My only suggestion is about the last section -- about addressing mental health issues. As the subtitle

stands now, it seems like the authors are going to talk about treatment of mental illness in jails and prisons. But it's really about how we account for mental illness as an explanation for criminal behavior. So maybe the subtitle for that section could be something like "redefining the legal concept of 'insanity'"?

On that note, would the authors consider adding a section about how psychology and neuroscience could inform treatment of serious mental illness (SMI) in jails and prisons? I understand if word-count limits don't allow for another section, but as a clinician I wonder how recent understandings of how mental illness is treated could inform how jails and prisons approach behavioral, cognitive, problem-solving, and medication treatment of SMI.

Reviewer #1 (Remarks to the Author):

The purpose of this commentary is to discuss deleterious impact of the criminal legal system and how psychology and neuroscience can reduce the footprint and consequences of the expansive criminal legal system in the United States. This is an interesting paper highlighting an important issue that is understudied and under-discussed (I've worked in corrections and been a correctional researcher for nearly 25 years and I cannot recall a paper I've read on the criminal legal footprint). The use of psychology and neuroscience certainly offers novel opportunities for combatting the impact of this footprint.

The strengths of this manuscript lie in the novelty of the topic and discussion of science, including neuroscience, to combat the negative impact of the criminal legal footprint. Limitations of the manuscript include instances of overstating results to make a point, and a lack of objectivity throughout the paper. I do see potential in this manuscript, but for this commentary to contribute to the scientific literature, a more objective and neutral position is warranted. Further, the direct recommendations and implications of how psychological science and neuroscience can positively alter current directions needs further development. As presented, the recommendations are overly vague and non-descript; thus, limiting the impact of the conclusions and recommendations.

Response: We are thankful to Reviewer 1 for noting that our approach was “novel.” We also appreciate the points Reviewer 1 made below and addressed each point by editing text and/or adding references. We completely agree with Reviewer 1 that presenting a balanced science on the relevant topics is key for building an empirical foundation that addresses the negative effects of the legal system on individuals and society at large. Based on this empirical foundation we provide *specific* recommendations at the end of each section (e.g., regulate and govern environmental standards given the scientific evidence of the negative effects of exposures to pollution on individuals, reduce focus on command and control policies that leave incarcerated people more stuck in prison environments and further compound the negative effects of those environments, limit used of solitary-use as a last resort and in line with the Mandela Rules based on the evidence from prisons and other settings that demonstrate the deleterious effect of solitary, redefine legal insanity and open up avenues for alternatives to incarcerating people with mental illnesses, of many kinds, based on the evidence of entrenched psychological and biological factors that undermine culpability). We also outline these recommendations in Figure 1. We also note that the goal of a perspectives piece is to “provide a forum for authors to discuss models and ideas from a personal viewpoint. They are more forward looking and/or speculative than Review articles and may take a narrower field of view. They may be opinionated but should remain balanced and are intended to stimulate discussion and new experimental approaches” (from the journal website). Therefore, in the revision we sought to provide more references to support the points, clarify when issues are “not agreed upon” in the literature, yet still advance a perspective that we hope stimulates conversation.

Specific suggestions for improving the manuscript are provided below:

1. On page 1, the authors write, “In most jails and prisons, there are well-documented violations...” I appreciate the citations, but does the evidence support this conclusion for “most jails and prisons?” Certainly, problems exist, but I’m not aware that the prevalence has been established and that it extends to most of the thousands of jails and prisons

across the United States. In fact, the paper goes on to counter this conclusion by citing that ¼ of California prisons have water contamination problems. This is disconcerting for sure, but does not represent “most” of the prisons in California, in fact, less than one-half. A more neutral position throughout the paper would increase the appearance of objectivity, and, thus, confidence in the conclusions.

Response: We appreciate that Reviewer 1 pointed out places where our language could be more precise. We edited the highlighted sentence.

Page 3: There are documented violations, ranging from inadequate sewage and waste disposal to poor water quality and the presence of toxins such as asbestos, manganese, and lead, in jails and prisons throughout the United States⁹⁻¹⁴.

We also reviewed the manuscript for other potential “overstatements.” We did not find other instances of language being used that exaggerated the numbers or inappropriately summarized the extant literature.

2. The review of impacts of incarceration, and all its variables, is superficial (p. 2-3). There is a body of work examining the impacts of incarceration on persons with mental illness that would allow for a more through discussion of the impact of jail/prison on individuals mental health functioning (including policy, culture, and environmental impacts).

Response: We completely agree with Reviewer 1 that the literature on the impact of incarceration is expansive. There are several topics one could tackle in a piece on incarceration. In the present manuscript, we selected three topics that are ripe for discussion in the context of empirical research from psychology and neuroscience. We believe it would be too much for the reader, and extend beyond the journal word limit, to add more topics. We also fear that we were unclear—our goal is not only to focus on people with mental illness. However, given the impact of exposure to toxins and solitary confinement on mental health, we comment on these outcomes in the related sections. Finally, given the goal of the manuscript, to think about ways to shrink the footprint of the legal system, we have one section focused on people with existing mental illness, in the context of our discussion on the legal conceptualization of insanity. We substantially edited the last paragraph of our introduction to reference the expansive literature noted by the reviewer and to better outline the scope of this piece.

Page 3: While there is much skepticism about the use of psychology and neuroscience in the legal system, these disciplines do have the potential to affect meaningful change in how the legal system operates and in the outcomes it produces. In this perspective piece, we will argue that psychological and neuroscientific findings can be applied to and improve aspects of the legal system through policy changes. We will focus on how science can shape policies that affect those who are incarcerated in jails and prisons, and by extension society at large. There is a substantial body of research delineating the negative impact of incarceration on individuals (e.g., negative effects on health, mental health, job prospects, educational attainment, etc.)³⁻⁶ and their families^{2,7}. Here, we select three aspects of where and who is incarcerated and detail how policies surrounding these aspects can or should be influenced by emerging findings in psychology and neuroscience. Specifically, we highlight how the issues of

pollution in prisons, the use of solitary confinement, and the restrictive legal concept of insanity could be reshaped by integrating scientific findings. Our aim is to lay out a premise for how such policy-based applications of science can bypass the constraints and requirements of both science and the law. Policies are not designed or implemented to affect just one individual. Much like research, policies are aimed at the group, or the population “on average.” Thus, aligning science and the law as applied to policy can serve the needs of many.

3. The discussion of solitary confinement is also superficial and does not take into account studies that demonstrate negligible effects. The literature reviewed is not inaccurate, but limiting the review to one perspective while ignoring a counter-perspective is incomplete at best, and biased at worst. The relevant points can be made with a more neutral review of the scientific literature.

Response: We appreciate Reviewer 1 flagging the few studies that demonstrate negligible effects of solitary confinement. We edited text on pages 5-6 to highlight the handful of studies that show null effects of solitary confinement on psychological health and security violations (citations 47-51). However, we also added additional studies demonstrating evidence that solitary confinement negatively impacts individuals (citations 49, 52-62). Further, we believe that the use of psychology and neuroscience work outside the context of prison to support the potential negative effects of solitary experiences provides additional evidence that this practice has the potential to make people worse than they were before incarceration and extend the effect of incarceration well-beyond the period of isolation.

Pages 5-6: Research on solitary confinement includes qualitative accounts of incarcerated persons’ experiences and empirical studies examining the relationship between this aspect of incarceration and safety, mental health, and criminogenic risk. While the qualitative accounts, as well as popular media sources and theory-based writings from scholars, document the harrowing effects of solitary on individuals^{45,46}, the empirical evidence supporting the negative effects of solitary on safety, mental health and criminogenic risk is more mixed. Some studies fail to detect effects of solitary confinement on individual behavior and mental health⁴⁷⁻⁵¹. Other studies document significant negative effects of solitary on incarcerated people’s physical and mental health⁵²⁻⁶¹, particularly in terms of anxiety psychotic symptoms, sensory arousal, and behaviors that effect mortality by any or unnatural causes (e.g., suicide)^{49,57}. Additionally, there is evidence that being housed in solitary confinement, even for a week, can change electrical activity in the brain^{49,62}. The U.S. Department of Justice acknowledges that solitary confinement can worsen existing mental illnesses and trigger new ones⁶³.

The study of solitary confinement is understandably very difficult. Some studies cited above lack appropriate methodological controls (e.g., randomization, comparison groups), were conducted in small samples, and/or were the result of litigation possibly introducing bias into the method^{64,65}. Unequivocal empirical evidence for concluding that the practice of solitary confinement in jails and prisons is uniformly negative is lacking, leading some scholars ^{47,49} to suggest caution in developing policy based on an incomplete science. However, there is a more substantial evidence base on the negative effects of solitary conditions in research with non-human animals and humans outside of the jail/prison context. While research in laboratories or in other

institutional setting is not identical to incarceration-based solitary, there is a strong basis for comparing the effects of physical isolation and the deprivation of basic experiences.

4. The use of psychopathy as the primary example of treating and managing persons with mental illness is problematic. Psychopathy is a personality disorder, and is clinically different from persons with severe mental illness (which represents, on average, 10-15% of the prison population, where as the prevalence rates for personality disorders are substantially higher). A better example would be a person with schizophrenia or bipolar disorder in jails and prisons (not insanity acquittals as they are patients in the mental health systems, not criminal legal systems). It would also be relevant to note, that the over-representation of people with mental illness is not merely a result of the deinstitutionalization movement as a relatively robust body of research demonstrates that justice-involved persons with mental illness present with criminal risk comparable to their non-mentally ill peers in the criminal legal system (see works by Drs. Jim Bonta, Jennifer Skeem, Jeffrey Drain, Robert Morgan, et al.).

Response: Both this reviewer and Reviewer 2 commented on the final section. The substantial changes we made were with both reviewers' comments in mind. First, we fear that our original naming of the final section was misleading. We did not intend to cover the vast topic of "treating and managing persons with mental illness." Rather, we aimed to focus on one legal concept- insanity - that we believe bloats the footprint of the legal system by deeming people who should receive substantive mental health care fit to be housed in traditional jails and prisons (often without appropriate services). Therefore, we renamed this section following Reviewer 2's suggestion "Redefining the Legal Concept of Insanity." Second, we completely revised our lead into the topic of insanity, clearly explaining what clinical psychology views as a mental illness (including personality disorders) versus how the law considers insanity (and restricts the definition of serious mental illness). We reference how the legal definition of insanity restricts the use of this type of defense of consideration in sentencing for *all* forms of mental illness. However, we stick with psychopathy as the case example because it is explicitly excluded in many legal doctrine, it is over-represented in the legal system (even compared to forms of SMI), and we want to challenge readers to ask why that is—we argue it is not in line with the science. We also added to our statistics on elevated rates of arrests among those with mental health problems a sentence that addresses the final point made by the reviewer. We provide references that include the scholars noted by the reviewer. Please see our extensive edits on pages 7-9.

5. The opinion that persons with psychopathy qualify for the insanity defense is not a policy or practice issue, it is a legal determination defined in federal and state statutes. This opinion is also an outlier opinion not well supported in the paper and is unlikely to be supported by the majority of mental health, legal, correctional or policy professionals that work in the criminal legal system. Again, an important concept not adequately covered in this opinion regarding psychopathy specifically and mental illness broadly is the prevalence of criminal risk.

Response: This submission is a perspective piece, which based on the journal instructions can "provide a forum for authors to discuss models and ideas from a personal viewpoint. They are more forward looking and/or speculative than Review articles and may take a narrower field of view. They may be opinionated but should remain balanced and are intended to stimulate discussion and new experimental

approaches.” Therefore, we are aware that the stated perspective on psychopathy is an outlier in the field and we hope to use the extant scientific basis to start a conversation about whether the current dominant view in psychology, neuroscience, and the law is appropriate or not. Disorders such as psychopathy are continually highlighted as related to risk for recidivism and crime (see Bonta et al, 2014, multiple writings by Jennifer Skeem, as well as the references cited in the submission). Despite this risk, we encourage readers to think about whether the science supports such as punitive approach to handling people with this disorder. We believe our edits outlining the logic of our argument have clarified that continuing to ignore the scientific evidence on psychopathy only serves to harm those with the disorder and undermines public safety. We also provide citations to comprehensive reviews that support the evidence presented about psychopathy on pages 7 and 8. The focus of this argument is consistent with the premise of the paper that psychological and neuroscientific evidence can be used to shape legal policies with an eye toward reducing the impact of the legal system. The policy of expanding the applicability of the insanity defense/consideration of insanity broadly defined (*nota bene*: lawyer on this submission, as well as others consulted in response to the review, believes is a legal policy) is one example of ways to reform current practices based on scientific evidence. In our revision, we do reference other forms of mental illness and provide citations to make clear we believe that *any* clinical disorder should be eligible for consideration under insanity doctrines.

Reviewer #2 (Remarks to the Author):

This is an interesting piece that turns the common narrative about psychology in the law (e.g., that psychologists can detect liars or predict crimes) into ways that psychological findings can be used to decrease harms caused by incarceration. The authors' recommendations are (or should be) common sense -- improve the quality of the jail/prison environment; reduce use of solitary confinement; revisit the definition of "insanity" to encompass a broader range of psychopathology. However, the piece does an impressive job of bringing together a concise summary of current and strong evidence from the literature to support each recommendation. This is more compelling than the argument that it's "just the right thing to do."

Response: We thank Reviewer 2 for commenting that our summary of the work was “impressive” and that we provide a “compelling” argument.

My only suggestion is about the last section -- about addressing mental health issues. As the subtitle stands now, it seems like the authors are going to talk about treatment of mental illness in jails and prisons. But it's really about how we account for mental illness as an explanation for criminal behavior. So maybe the subtitle for that section could be something like "redefining the legal concept of 'insanity'"?

Response: We so appreciate Reviewer 2's suggestion for renaming the last section. We followed this suggestion, and the last section is now named: “Redefining the Legal Concept of Insanity” (page 7).

On that note, would the authors consider adding a section about how psychology and neuroscience could inform treatment of serious mental illness (SMI) in jails and prisons? I understand if word-count limits don't allow for another section, but as a clinician I wonder how recent understandings of how mental illness is treated could inform how

jails and prisons approach behavioral, cognitive, problem-solving, and medication treatment of SMI.

Response: We revised this section (page 7) to expand on the points made on the handling of serious mental illness (SMI) and available treatments based on current psychological and neuroscientific evidence (page 8). However, we did not go into detail about SMI versus personality disorders due to word limits and wanting to keep our discussion focused on what the law defines as insanity and how personality disorders, such as psychopathy, are excluded. We ask the question “does the scientific evidence support such a legal practice or is society blinded by its moralistic outrage at the unfathomably cruel behavior these individuals? (page 8). A goal of this perspectives piece is to discuss ways we can reduce the impact of the legal system. Incarcerating a large number of individuals (i.e., those with psychopathy) who have documented psychological and neural atypicalities that impact basic information processing and reasoning expands the footprint of the legal system beyond what is scientifically supported. Instead, consistent with the point made by Reviewer 1 (e.g., “patients in the mental health systems, not criminal legal systems”) we argue that a shift in the application of the insanity defense open up avenues for where people with *different* forms of mental illness can be treated. To be responsive to this review, we did add several references (^{98,99, 101,102,110,111}) and sentences (pages 8 and 9) to the section on SMI generally and the treatability of these disorders.

1st Feb 24

Dear Arielle,

Thank you for your patience during the peer-review process. Your manuscript titled "Shrinking the Footprint of the Criminal Legal System through Policies Informed by Psychology and Neuroscience" has now been seen by the same 2 reviewers as before, and I include their comments at the end of this message.

The reviewers find your work significantly improved, and we agree with this assessment. At the same time, we also share Reviewer #1's serious concerns regarding the discussion and proposals relating to mental health, especially as included in the section "Redefining the Legal Concept of Insanity". One aspect that we consider especially problematic is that the proposal presumes that people with mental illness are justice-involved as a direct result of their mental illness (for which substantive evidence is lacking); while it is correct that Perspectives may be more forward-looking than Reviews, each argument must be empirically grounded, with the empirical strength of evidence ascertained by experts in the field. Here, the argument is rhetorically grounded and too speculative for the format. To consider a revised version of the manuscript, we would need the entire section to be rewritten or removed, and in particular not refer to ASPD or psychopathy as the exemplified diagnoses.

We appreciate that these requests for revisions are major and may make you reconsider whether you wish to put forward your proposal as a Perspective in Communications Psychology. We remain very interested in the possibility of publishing your manuscript in Communications Psychology, but understand if you withdraw from the consideration process in light of the request for further substantive revisions.

Please use the following link to submit your revised manuscript and a point-by-point response to the referees' comments (which should be in a separate document to any cover letter):

[Link redacted]

We hope to receive your revised paper within 12 weeks; please let us know if you aren't able to submit it within this time so that we can discuss how best to proceed. If we don't hear from you, and the revision process takes significantly longer, we may close your file.

If you decide to withdraw the manuscript, we would be grateful if you could let us know so that we can close your file. Otherwise, we look forward to seeing the revised manuscript and - in any case - thank you for the opportunity to review your work.

Best wishes,

Marika

Marika Schiffer, PhD
Chief Editor
Communications Psychology

REVIEWERS' COMMENTS:

Reviewer #1 (Remarks to the Author):

I commend the authors for addressing the reviewer comments. It is clear to me that the authors took great care to address each critique, and the result is an improved paper. Although I believe the paper is much improved, I remain unconvinced regarding the section entitled "Redefining the Legal Concept of Insanity." First, this section seems to adopt the premise that people with mental illness are justice-involved as a direct result of their mental illness, and we know this to not be the case (see previously recommended works of Jennifer Skeem, Jeffrey Drain, and Robert Morgan). In fact, Skeem and colleagues estimate that only about 10% of people with mental illness that are justice involved are so involved as a direct result of their illness – that is, approximately 90% of justice-involved persons with mental illness appear to be justice-involved for reasons other than mental illness – reasons of criminal risk like their non-mentally ill peers. This paper does not recognize this important distinction. The fact is, the majority of people with mental illness that are justice-involved are so involved because they present with substantive criminal risk – they think, feel, and act in ways that are consistent with people that are justice-involved but not suffering from a mental illness.

Secondly, the notion that people with antisocial personality disorder (ASPD) or psychopathy be diverted from the legal system to the mental health system is illogical. The equivalent would be to say that people with mental illness that break the law as a direct result of their illness should be diverted from the mental health system and to the legal system merely by the fact that they committed a crime. Both are equally illogical and dangerous. The diagnosis of ASPD or psychopathy, by definition, is someone that is going to violate the rights of others/the law, that is exactly who we aim to involve in the justice system (and potentially incarcerate) as a means of public safety. As written, I do not see a compelling argument for the authors contention that these people, simply by meeting the criteria for a diagnosis, should qualify for the insanity defense. In fact, most estimates of the prevalence of people with ASPD in the justice system is much higher than the authors cited 50% (usually more like 70-75%). This is not an over-prevalence of people with ASPD in our jails and prisons, rather, our jails and prisons were built for people that are unable to control or manage their behavior and thus harm others. The fact that we have developed a mental health diagnosis for these individuals does not (and should not) alter their legal status. I appreciate that the authors were responsive to reviewer critiques regarding this section; however, as presented, I find this section, which is a substantive portion of the paper, to be illogical and not reasonably grounded in science. It is an extreme perspective that does not fit with policy or science related to the issue of insanity.

I do appreciate the authors note that "the goal of a perspectives piece is to provide a forum for authors to discuss models and ideas from a personal viewpoint. They are more forward looking and/or speculative than Review articles and may take a narrower field of view. They may be opinionated but should remain balanced and are intended to stimulate discussion and new experimental approaches" (from the journal website)...yet still advance a perspective that we hope stimulates conversation." These allowances noted, such papers are still expected to maintain a tie to science and do not provide an open forum for illogical and extremist positions under the guise of "creating thought." I maintain my initial opinion that the author's point in this third section of the paper can still be made without the use of ASPD or psychopathy as the exemplified diagnosis. Using these diagnoses diminishes the focus of the section, is unnecessarily extreme, and quite frankly, hinders the potential impact of the paper.

Reviewer #2 (Remarks to the Author):

The authors have addressed my concerns and I have no further concerns.

Reviewer #1 (Remarks to the Author):

I commend the authors for addressing the reviewer comments. It is clear to me that the authors took great care to address each critique, and the result is an improved paper.

Response: We appreciate the Reviewer's positive remarks.

Although I believe the paper is much improved, I remain unconvinced regarding the section entitled "Redefining the Legal Concept of Insanity." First, this section seems to adopt the premise that people with mental illness are justice-involved as a direct result of their mental illness, and we know this to not be the case (see previously recommended works of Jennifer Skeem, Jeffrey Drain, and Robert Morgan). In fact, Skeem and colleagues estimate that only about 10% of people with mental illness that are justice involved are so involved as a direct result of their illness – that is, approximately 90% of justice-involved persons with mental illness appear to be justice-involved for reasons other than mental illness – reasons of criminal risk like their non-mentally ill peers. This paper does not recognize this important distinction. The fact is, the majority of people with mental illness that are justice-involved are so involved because they present with substantive criminal risk – they think, feel, and act in ways that are consistent with people that are justice-involved but not suffering from a mental illness.

Response: We fear that there is misinterpretation of our points in this section. Based on this review and comments from the Editor we substantially revised the section (pages 7-8). Instead of speaking to a specific disorder we invite readers to consider how the misalignment between the legal doctrines and psychology/neuroscience might stifle progress toward reducing the footprint of the legal system (i.e., incarcerating people in traditional prisons who might have mitigating factors that would deem them "insane"). Now, we do not take a position on whether people with certain mental health problems should or should not be deemed as "insane" based on the law, but we encourage a shift toward allowing people to put forth that defense and have the courts decide if that is an appropriate justification for the behavior. We absolutely agree that not everyone with a mental health problem can use that as an excuse/mitigating factor in their behavior. We simply believe based on the science and the words of the law that there should not be an arbitrary restriction.

Secondly, the notion that people with antisocial personality disorder (ASPD) or psychopathy be diverted from the legal system to the mental health system is illogical. The equivalent would be to say that people with mental illness that break the law as a direct result of their illness should be diverted from the mental health system and to the legal system merely by the fact that they committed a crime. Both are equally illogical and dangerous. The diagnosis of ASPD or psychopathy, by definition, is someone that is going to violate the rights of others/the law, that is exactly who we aim to involve in the justice system (and potentially incarcerate) as a means of public safety. As written, I do not see a compelling argument for the authors contention that these people, simply by meeting the criteria for a diagnosis, should qualify for the insanity defense. In fact, most estimates of the prevalence of people with ASPD in the justice system is much higher than the authors cited 50% (usually more like 70-75%). This is not an over-prevalence of people with ASPD in our jails and prisons, rather, our jails and prisons were built for people that are unable to control or manage their behavior and thus harm others. The fact that we have developed a mental health diagnosis for these individuals does not (and should not) alter their legal status. I appreciate that the authors were responsive to

reviewer critiques regarding this section; however, as presented, I find this section, which is a substantive portion of the paper, to be illogical and not reasonably grounded in science. It is an extreme perspective that does not fit with policy or science related to the issue of insanity.

Response: Rather than getting into a tit-for-tat about each of the points made above, we will note that we revised the section. We are very clear that our focus on revising the insanity defense application is not to remove consequence but to appropriately acknowledge the potential mitigating factors that *may* exist for an individual and use that defense to potentially route someone to a forensic mental health center (if successful in their defense) rather than a traditional prison. Our argument is not that the people with severe mental illness do not or cannot pose public safety risks, but rather that imprisonment has a well-documented criminogenic effect that undermines public safety, including by exacerbating underlying mental health contributors to criminal behavior and failing to remediate them. Our point is not that people with mental illness should receive some sort of automatic diversion to the mental health system, but rather that the standards by which the legal system evaluates insanity and culpability unduly narrows the scope of who gets assessed for insanity-based defenses, which in turn precludes the court from determining an appropriate response (including forensic mental health treatment, if warranted).

I do appreciate the authors note that “the goal of a perspectives piece is to provide a forum for authors to discuss models and ideas from a personal viewpoint. They are more forward looking and/or speculative than Review articles and may take a narrower field of view. They may be opinionated but should remain balanced and are intended to stimulate discussion and new experimental approaches” (from the journal website)...yet still advance a perspective that we hope stimulates conversation.” These allowances noted, such papers are still expected to maintain a tie to science and do not provide an open forum for illogical and extremist positions under the guise of “creating thought.” I maintain my initial opinion that the author’s point in this third section of the paper can still be made without the use of ASPD or psychopathy as the exemplified diagnosis. Using these diagnoses diminishes the focus of the section, is unnecessarily extreme, and quite frankly, hinders the potential impact of the paper.

Response: We removed the focus on psychopathy. We made the general points outlined above without reference to a specific disorder.

Reviewer #2 (Remarks to the Author):

The authors have addressed my concerns and I have no further concerns.

Response: We appreciate the Reviewer’s endorsement.

26th Mar 24

Dear Arielle,

Your Perspective titled "Shrinking the Footprint of the Criminal Legal System through Policies Informed by Psychology and Neuroscience" has now been seen by Reviewer #1, whose comments appear below. In the light of their advice I am delighted to say that we are happy, in principle, to publish it in Communications Psychology under a Creative Commons 'CC BY' open access license.

If the revised paper is in Communications Psychology format, in accessible style and of appropriate length, we shall accept it for publication immediately. I have attached an edited version of your manuscript, and ask you to attend to each comment in detail.

EDITORIAL REQUESTS:

* Please review the changes in the attached copy of your manuscript, which has been edited for style, and address the comments and queries I have added. If using Word, please use the 'track changes' feature to make the process of accepting your manuscript more efficient.

* Please check whether your manuscript contains third-party images, such as figures from the literature, stock photos, clip art or commercial satellite and map data. If any of the display items in your manuscript (figures, tables, boxes or movies) include images that are the same as, or are adaptations of, previously published images, please fill in the Third Party Rights Table, and return to us when you submit your revised manuscript. This information will enable us to obtain the necessary rights to re-use such material. If we are unable to obtain the necessary rights to use or adapt any of the material that you wish to use, we will contact you to discuss alternative options.

* Communications Psychology uses a transparent peer review system. On author request, confidential information and data can be removed from the published reviewer reports and rebuttal letters prior to publication. If you are concerned about the release of confidential data, please let us know specifically what information you would like to have removed. Please note that we cannot incorporate redactions for any other reasons.

*If you have not done so already, please alert me to any related manuscripts from your group that are under consideration or in press at other journals, or are being written up for submission to other journals (see www.nature.com/authors/editorial_policies/duplicate.html for details).

FORMATTING GUIDELINES:

You will find a complete list of formatting requirements following this link:

<https://www.nature.com/documents/commsj-style-formatting-checklist-review-perspective.pdf>

Please use the checklist to prepare your manuscript for final submission. In the following, I also highlight some issues of particular importance.

** Figures

Please upload Figures individually, one figure per file. To ensure the swift processing of your paper please provide the highest quality, vector format, versions of your images (.ai, .eps, .psd) where available. Text and labelling should be in a separate layer to enable editing during the production process. If vector files are not available then please supply the figures in whichever format they were compiled in and not saved as flat .jpeg or .TIFF files. If your artwork contains any photographic

images, please ensure these are at least 300 dpi.

* References

References appear as superscript Arabic numerals, in order of mention. The reference list mentions references in the numerical order in which they are mentioned in the main text. If a reference is cited more than once, the same number is used throughout the text and the reference receives a single entry in the reference list.

We ask that you select the most significant 5–10% of references in your list for highlighting, and add a single sentence in bold after each of these references to describe the main result and its significance.

Only papers that have been published or accepted by a named publication should be in the reference list (preprints and citations of datasets are also permitted). Unpublished/Submitted research should not be included in the reference list; it should only be mentioned briefly and parenthetically in the main text. Note that no major arguments should rely on unpublished research.

Published conference abstracts and URLs for web sites should be cited parenthetically in the text, not in the reference list.

Footnotes are not used.

* Competing interests

Please include a "Competing interests" statement after the References. Note that we ask authors to declare both financial and non-financial competing interests. For more details, see <https://www.nature.com/authors/policies/competing.html>. If you have no financial or non-financial competing interests, please state so: "The authors declare no competing interests."

SUBMISSION INFORMATION:

* Your paper will be accompanied by a two-sentence editor's summary, of between 250-300 characters, when it is published on our homepage. Could you please approve the draft summary below or provide us with a suitably edited version.

This Perspective calls for a reform of the criminal justice system in the US. Psychological and neuroscientific research should inform regulations around pollution and toxins, policies for solitary confinement, and the framework for the admissibility of legal insanity defence.

In order to accept your paper, we require the following:

* A cover letter describing your response to our editorial requests.

* The final version of your text as a Word or TeX/LaTeX file, with any tables prepared using the Table menu in Word or the table environment in TeX/LaTeX and using the 'track changes' feature in Word.

* Production-quality versions of all figures, supplied as separate files. Photographic images should be 300 dpi in RGB format (.jpg, TIFF or native Photoshop format) and any labels/scale bars included in a

separate layer from the image. Line art, graphs and schemes should be vector format (.ai, .eps, .pdf); Adobe Illustrator files are preferred and will minimize production time. Any chemical structures or schemes contained within figures should additionally be supplied as separate Chemdraw (.cdx) files.

At acceptance, the corresponding author will be required to complete an Open Access Licence to Publish on behalf of all authors, declare that all required third party permissions have been obtained.

Please note that your paper cannot be sent for typesetting to our production team until we have received this information; **therefore, please ensure that you have this ready when submitting the final version of your manuscript.**

ORCID

Communications Psychology is committed to improving transparency in authorship. As part of our efforts in this direction, we are now requesting that all authors identified as 'corresponding author' create and link their Open Researcher and Contributor Identifier (ORCID) with their account on the Manuscript Tracking System (MTS) prior to acceptance. ORCID helps the scientific community achieve unambiguous attribution of all scholarly contributions. For more information please visit <http://www.springernature.com/orcid>

For all corresponding authors listed on the manuscript, please follow the instructions in the link below to link your ORCID to your account on our MTS before submitting the final version of the manuscript. If you do not yet have an ORCID you will be able to create one in minutes.

IMPORTANT: All authors identified as 'corresponding author' on the manuscript must follow these instructions. Non-corresponding authors do not have to link their ORCIDs but are encouraged to do so. Please note that it will not be possible to add/modify ORCIDs at proof. Thus, if they wish to have their ORCID added to the paper they must also follow the above procedure prior to acceptance.

To support ORCID's aims, we only allow a single ORCID identifier to be attached to one account. If you have any issues attaching an ORCID identifier to your MTS account, please contact the Platform Support Helpdesk.

[Link redacted]

We hope to hear from you within two weeks; please let us know if the process may take longer.

Best regards,

Marike

Marike Schiffer, PhD
Chief Editor

Communications Psychology

REVIEWERS' COMMENTS:

Reviewer #1 (Remarks to the Author):

I appreciate the authors responsiveness to my feedback. I believe the revised manuscript communicates clearly the points the authors intend to make, and as revised, can make a substantial contribution to the literature. I have no additional recommendations for this manuscript.